# Colonization of methicillin-resistant *Staphylococcus aureus* and vancomycin-resistant *Enterococci* and its associated factors in cancer patients at the University of Gondar Comprehensive Specialized Hospital, Northwest Ethiopia

**Eden Getaneh Mekonnen**[1]*, **Abebe Birhanu**[1], **Mulugeta Yimer**[2], **Segenet Bizuneh**[3], **Mucheye Gizachew**[1], **Baye Gelaw**[1]

1 Department of Medical Microbiology, School of Biomedical and Laboratory Sciences, College of Medicine and Health Sciences, University of Gondar, Northwest Ethiopia, 2 Department of Pediatrics and Child Health, School of Medicine, College of Medicine and Health Sciences, University of Gondar, Northwest Ethiopia, 3 Department of Internal Medicine, School of Medicine, College of Medicine and Health Sciences, University of Gondar, Northwest Ethiopia

* edenget48@gmail.com, eden.getaneh@uog.edu.et

## Abstract

### Background

Cancer patients are predisposed to methicillin-resistant *Staphylococcus aureus* and vancomycin-resistant *Enterococci* colonization. However, the prevalence of these pathogens among cancer cases in Northwestern Ethiopia remains underreported.

### Objective

To determine the prevalence of colonization of methicillin-resistant *Staphylococcus aureus* and vancomycin-resistant *Enterococci* and associated factors among cancer patients at the University of Gondar Comprehensive Specialized Hospital, Northwestern Ethiopia.

### Method

A cross-sectional study enrolled 288 confirmed cancer participants through stratified systematic random sampling, gathering socio-demographic and clinical data via pretested structured questionnaires from May 1 to July 30, 2023. Each participant provided two specimens: a nasal swab and a fecal sample. Nasal swabs were collected using sterile swabs, inserted at least 1 cm into each nostril, and rotated against the nasal membrane for 10 to 15 seconds, which were then placed in Amies transport medium. Fecal specimens were collected in leak-proof plastic containers, swabbed, and transferred to Cary Blair transport medium. Nasal swabs and fecal specimens were cultured on Mannitol salt agar at 37°C for *Staphylococcus aureus* identification, which was confirmed by coagulase

**Data availability statement:** All relevant data is within the manuscript and its supporting information files.

**Funding:** The author(s) received no specific funding for this work.

**Competing interests:** The authors have declared that no competing interests exist.

**List of abbreviation:** ANC, Absolute Neutrophil Count; CLSI, Clinical Laboratory Standards Institute; MIC, Minimum Inhibition Concentration; MDR, Multi Drug Resistance; MHA, Muller Hinton Agar; MRSA, Methicillin Resistant *Staphylococcus aureus*; MSA, Mannitol salt agar; MSSA, Methicillin Susceptible Staphylococcus aureus; UoGCSH, University of Gondar Comprehensive Specialized Hospital; VRE, Vancomycin Resistant Enterococci; VRSA, Vancomycin Resistant *Staphylococcus aureus*; WBC, White Blood cell Count.

testing and Gram staining. *Enterococci* were cultured on Bile esculin agar at 43°C and identified at the genus level by cultural characteristics, with confirmation through Gram reaction and catalase tests. Antibiotic susceptibility was evaluated using the Kirby-Bauer disk diffusion method, with minimum inhibitory concentrations for vancomycin determined via E-test strips. To detect methicillin-resistant *Staphylococcus aureus*, a cefoxitin disk was used. Inducible clindamycin resistance in *Staphylococcus aureus* was determined by the D test. Epi-info version 7 and SPSS version 27 were used for data entry and data analysis, respectively. The Pearson Chi-Square test was initially used to evaluate the association between factors and outcomes as the preliminary analysis, with a significance threshold of $p < 0.05$. Variables meeting this criterion underwent bivariable and multivariable logistic regression analyses, using p-value cutoffs of $< 0.2$ for bivariable and $< 0.05$ for multivariable analyses.

## Result

The study involved 288 participants, with 51.0% being men and a mean age of 45.6 years. The prevalence of methicillin-resistant *Staphylococcus aureus* was 11.1% (95% CI: 7.5–14.7%), while vancomycin-resistant *Enterococci* had a prevalence of 2.8% (95% CI: 0.9–4.7%). Inducible clindamycin-resistant *Staphylococcus aureus* comprised 13.5% of the isolates. The multidrug-resistant proportion of *Staphylococcus aureus* and *Enterococci* were 56.2% and 55.2%, respectively. Both organisms exhibited the highest resistance to the antibiotic classes of penicillin and tetracycline. Significant associations were identified between methicillin-resistant *Staphylococcus aureus* colonization and low absolute neutrophil count (AOR = 13.050, 95% CI: 1.362-125.00, P = 0.026), and between vancomycin-resistant *Enterococci* colonization and having undergone an invasive procedure (AOR = 8.648, 95% CI: 1.870-39.992, P = 0.006).

## Conclusion

The study reveals a significant prevalence of methicillin-resistant *Staphylococcus aureus* and vancomycin-resistant *Enterococci* colonization among cancer patients, raising public health concerns. High antibiotic resistance rates complicate treatment and may impact patient outcomes. Notably, the high inducible clindamycin resistance report, highlights the need for D-testing. Screening for methicillin-resistant *Staphylococcus aureus* is recommended as an important antibiotic stewardship measure, while early detection of vancomycin-resistant *Enterococci* colonization is crucial to reduce complications.

## Introduction

Cancer patients are defined by the uncontrolled proliferation and dissemination of abnormal cells throughout the body [1]. They are at increased risk of infections and colonization due to factors like chemotherapy, neutropenia, central venous catheters, frequent antimicrobial use, hospitalizations, and surgical interventions [2,3].

 Methicillin-resistant *Staphylococcus aureus* (MRSA) is a strain of *Staphylococcus aureus* (*S. aureus*) which is resistant to a large group of antibiotics called the beta-lactams [4]. It is a significant human pathogen characterized by its production of various toxins, including

alpha-toxin and Panton-Valentine leukocidin, which contribute to tissue damage and immune evasion [5]. These virulence factors, combined with MRSA's ability to form biofilms, enhance its persistence on medical devices and within host tissues, complicating treatment efforts [6].

*Enterococci* are gram-positive bacteria that occur as part of the normal flora in the gastro-intestinal (GI) tract of humans and animals [7]. Vancomycin-resistant *Enterococci* (VRE) are antibiotic-resistant bacteria commonly found in healthcare settings. They primarily employ surface adhesins as their main virulence factors, facilitating colonization of host tissues, especially in the GI tract [8]. Additionally, VRE can create biofilms that are resistant to immune responses and antibiotics, enabling their persistence on medical devices and within the body [9].

Methicillin-resistant *S. aureus* and VRE pose significant challenges in antibiotic resistance through various mechanisms. Methicillin-resistant *S. aureus* primarily resists beta-lactam antibiotics via the mecA gene, which produces an altered penicillin-binding protein with low affinity for these drugs [10]. It uses efflux pumps to expel antibiotics, forms biofilms for drug protection, produces beta-lactamase enzymes, and exhibits genetic plasticity for rapid resistance gene acquisition [2]. Similarly, VRE acquires resistance primarily through the vanA and vanB genes, which modify peptidoglycan precursors to diminish vancomycin's effectiveness [7]. Drug-resistant *Enterococci* use efflux pumps and protective biofilms to evade antibiotic treatment, often associated with specific genes like vanA and vanB, which enable the bacteria to acquire resistance through horizontal gene transfer, commonly involving mobile genetic elements on plasmids [11].

Methicillin-resistant *S. aureus* infections are linked to severe outcomes, including pneumonia, endocarditis, and skin infections, particularly in vulnerable populations such as the elderly and immunocompromised [12]. Similarly, VRE infections can lead to severe complications, including bloodstream infections, which have a mortality rate of up to 50% in critically ill patients [13]. The prevalence of MRSA and VRE in Ethiopia is notable, with a systematic review indicating a pooled MRSA nasal colonization prevalence of 10.94%. Regional variations exist, with the Oromia region having the highest prevalence at 21.28%, followed by the Amhara region at 6.78% [14]. Additionally, a recent study found a 6.5% prevalence of MRSA colonization among cancer patients [15]. For VRE, the pooled prevalence is estimated at 14.8%, with increasing rates of multidrug resistance among *Enterococci* isolates [16], although specific data on VRE colonization in the current study population is lacking.

Methicillin-resistant *S. aureus* colonization in the nares increases the risk of MRSA infection [17]. Its colonization in cancer patients, especially those with febrile neutropenia, increases infection risk and requires appropriate antibiotic therapy to prevent adverse clinical outcomes [18]. The increase in MRSA colonization in acute care facilities is largely due to patient-to-patient transmission, often through contaminated hands, clothes, or equipment of healthcare workers [19]. Colonization with VRE is particularly hazardous for cancer patients, as chemotherapy-induced mucosal damage may facilitate pathogen entry into the bloodstream, thereby increasing their clinical burden [20]. Moreover, immunosuppression and invasive procedures like urinary and vascular catheters are known risk factors for VRE colonization [21]. It also pose a growing medical concern due to their increasing prevalence and ability to spread vancomycin resistance to other bacteria, including *S. aureus* [22,23]. High-level vancomycin resistance in *Enterococci*, with the potential to transfer to *S. aureus*, jeopardizes the effectiveness of vancomycin for treating multidrug-resistant infections [24].

Vancomycin and Teicoplanin are commonly used to treat MRSA infections; however, due to glycopeptide resistance, alternative therapies such as Macrolide (erythromycin), Lincos-amide (clindamycin), and Streptogramin B (quinupristin-dalfopristin) antibiotic agents (collectively, MLSB agents) have been recommended [25]. Out of the MLSB agents, Clindamycin

is the most commonly used and preferred antibiotic due to its good oral absorption and tissue penetration. However, excessive use has led to increased resistance to these antibiotics. This inducible clindamycin resistance can be identified by the D-test [26,27].

Multi-drug resistance (MDR) is defined as acquired non-susceptibility to at least one agent in three or more antibiotic categories [28]. Multidrug resistance in MRSA and VRE poses significant challenges in healthcare settings globally [29]. To address the growing threat posed by MRSA and VRE in clinical settings, it is essential to comprehend and combat antibiotic resistance in these organisms. Understanding colonization can aid in the treatment of cancer patients and mitigate the potential infection effects in immunocompromised individuals. To the best of our knowledge, there are no study reports on the prevalence of colonization of MRSA and VRE studied together among cancer patients in the study area and Ethiopia.

## Materials and methods

### Study design, period, and setting

An institution-based cross-sectional study was carried out at the University of Gondar Comprehensive Specialized Hospital (UoGCSH) Oncology Treatment Center from May 1 to July 30, 2023. The hospital is located in the city of Gondar, 182 km from Bahir Dar, the capital city of the Amhara National Regional State, and 747 km from Addis Ababa, the capital city of Ethiopia. The UoGCSH is one of the largest tertiary referral and teaching hospitals in the region, serving over 13 million residents in northwestern Ethiopia. Among other services, the established cancer treatment center provides services for cancer cases. Currently, UoGCSH has separate pediatric and adult oncology units, treating nearly 3,000 cancer patients annually. The cancer treatment center includes outpatient and inpatient departments, each with 40 beds for pediatric and adult oncology [30].

### Source and study population

**Source population.** All patients who visited the UoGCSH during the study period were the source population.

**Study population.** The study population consisted of patients who visited the UoGCSH oncology treatment center for diagnosis and treatment from May 1 to July 30, 2023.

### Sample size determination and sampling technique

The sample size was calculated using a single population proportion formula, factoring in a 5% margin of error, a 95% confidence interval (Za/2 = 1.96).

The current study had two objectives

1. Prevalence of MRSA colonization

   For this particular objective, A 2020 study investigated MRSA colonization in cancer patients in the study area that reported an MRSA colonization prevalence of 6.5% [15].

2. Prevalence of VRE colonization

   This objective lacks prior studies on the same population, so a p-value of 0.5 was selected to increase the required sample size, a common practice in the absence of previous research.

   Because the study has two objectives, the one with the higher p-value typically determines the sample size. Therefore, we selected the second objective with a prevalence of 0.5 for our calculations.

   With an annual patient expectation of under 10,000 at the UoGCSH oncology treatment center, the sample size was adjusted to 261. Accounting for a 10% nonresponse rate, the total sample

size was adjusted to approximately 288. Data records at the UoGCSH showed that a total of 800 patients visited the hospital as oncology case within the last 3 months. We guessed that the study would take 3 months by considering 22 working days per a month. Therefore, 800 patients were expected to visit the UoGCSH oncology treatment center during the study period.

## Stratified systematic random sampling was used to include study participants as

**Pediatric oncology inpatient.** Cancer patients aged ≤ 18 years admitted to the children's ward for chemotherapy; N.B:- The UoGCSH Pediatrics oncology ward does not have a proper OPD so the all the pediatrics study participants were included were inpatients.

**Adult oncology inpatient.** Cancer patients aged > 18 years admitted to the adult oncology ward for chemotherapy

**Adult oncology OPD.** cancer patients aged > 18 years receiving follow-up and routine examinations without undergoing chemotherapy

A systematic random sampling method was used to recruit study participants from the above-mentioned stratum.

Probability proportional to size/PPS (proportional allocation technique) was obtained by using the formula Nf = (ni/N * n) = Nf = Average no of each ward patients × total sample size.

Total admitted patients in all wards

Pediatrics Inpatient = 83 x 288= 29.8 ≈ 3 0
800

Adult Inpatient ward n =390 x 288 = 140
800

Adult Outpatient n=327 x 288 =117.7 ≈ 118
800

Systematic random sampling was employed to select study participants by calculating the kth value, with a total population (N) of 800 based on data from the previous three months of hospital admissions: adult OPD (140), adult inpatient (118), and pediatric inpatient (30) (Fig 1). From May 1 to July 30, 2023, participants were recruited from the UoGCSH oncology treatment center using this method, including every kth value. Kth value of Adult OPD = 327/118 = 2.7 ≈ 3, Kth value of Adult inpatient = 390/140 = 2.78 ≈ 3, Kth value of Pediatrics inpatient = 83/30 ≈ 2.7 ≈ 3.

## Inclusion and exclusion criteria

All cancer-confirmed patients visiting the UoGCSH oncology treatment center for diagnosis and treatment between May 1 and July 30, 2023, were eligible for the study. Participants who were on antibiotic therapy during the data collection period, those who had an upper respiratory tract infection or symptoms such as coughing or difficulty breathing in the three months prior, and participants who did not have updated whole blood cell reports were excluded from the study.

## Variables of the study

The dependent variables in the present study include the prevalence of MRSA and VRE colonization among cancer patients. The independent variables comprise socio-demographic factors (sex, age, residence, marital status, education, and occupation status), cancer type, stage, chemotherapy cycles, comorbidities, antibiotic use, hospital stay duration, invasive procedures, chemotherapy treatment, presence of invasive procedures, white blood cell count (WBC), and absolute neutrophil count (ANC).

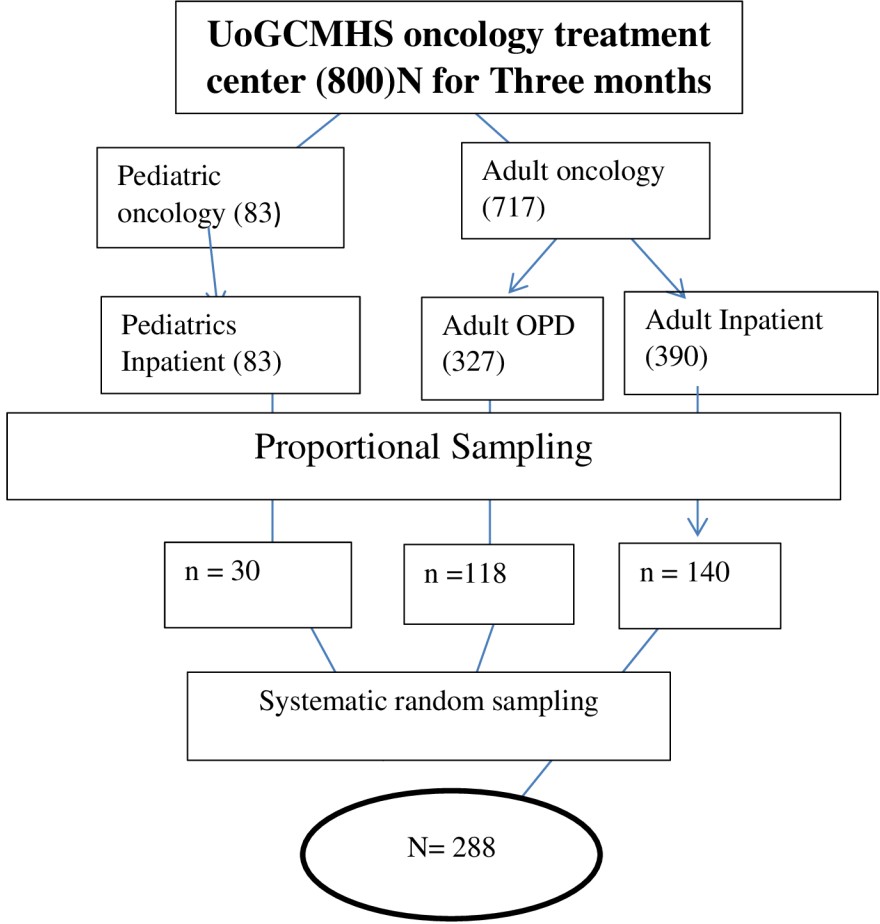

**Fig 1. Sampling procedure: the diagrammatic presentation of the sampling procedure of patients at UoGCSH oncology treatment center, Northwest Ethiopia.**

## Data collection and laboratory methods

**Socio-demographic data collection.** Socio-demographic data were collected through a structured, pre-tested questionnaire administered by an interviewer.

**Specimen collection and transportation.** Each study participant provided two specimens: a nasal swab and a fecal specimen. All specimen containers were labeled with a unique specimen number, date, type, and time of collection. Nasal swabs were collected with sterile swabs, inserted at least 1 cm into the nostril, and rotated against the nasal membrane for 10 to 15 seconds. Both nostrils of each participant were sampled with the same swab, which was then placed in Amies transport medium (HiMedia; Mumbai, India). Fecal specimens were collected in leak-proof plastic containers, swabbed, and transferred to Cary Blair transport medium (Oxoid; Hampshire, UK). Processing and culture were performed at the Medical Microbiology Teaching Laboratory of the College of Medicine and Health Sciences, University of Gondar.

## Identification of *Staphylococcus aureus* and *Enterococci* species

To identify *S. aureus*, a clinical specimen was inoculated onto Mannitol salt agar (MSA) (HiMedia; Mumbai, India) plates, which were then incubated at 37 °C for 24 hours. The

plates were examined for growth, with yellow colonies indicating successful mannitol fermentation, a characteristic feature of *S. aureus* [15]. Subsequently, a Gram stain was performed on selected young colonies, and Gram-positive cocci in clusters were noted, while the catalase test was then conducted by transferring a small amount of the isolated colony onto a microscope slide, and a drop of hydrogen peroxide was added. The observation of bubbling was indicative of a positive catalase reaction suggestive of *Staphylococcus* species [31]. A colony was then inoculated into sheep blood plasma for a coagulase test, with a slide coagulation test. A clot presence indicated a positive result, confirming the identification of *S. aureus* [32].

*Enterococci* identification was done by inoculating a specimen onto Bile esculin agar (Microxpress; Goa, India) and incubating at 43 °C for 24 hours. Black-pigmented colonies indicated esculin hydrolysis, characteristic of *Enterococci*. Selected colonies with distinct traits were sub cultured to obtain pure colonies [33]. A single black colony on Bile esculin agar (Microxpress; Goa, India) was gram-stained, showing gram-positive cocci typically in pairs or short chains. A catalase test followed, and the absence of bubbling confirmed a negative reaction, consistent with *Enterococci* species [34].

## Antibiotic susceptibility test

Disc diffusion and E-test minimum inhibitory concentration (MIC) for vancomycin (glycopeptide) (BioMérieux; Marcy-l'Étoile, France) methods were used to test for antibiotic susceptibility patterns. The concentrations of the antibiotics were based on the Clinical and laboratory standards institute (CLSI) 2022 recommendations [35]. The antibiotic disks (Oxoid; Hampshire, UK) tested for *S. aureus* were penicillin (penicillins) (10 units), erythromycin (macrolide) (15 µg), ciprofloxacin (fluoroquinolone) (5 µg), clindamycin (lincosamides) (2 µg), trimethoprim-sulfamethoxazole (sulfonamide) (25 µg), tetracycline (tetracyclines) (30 µg), gentamicin (aminoglycoside) (10 µg), chloramphenicol (phenicols) (30 µg), nitrofurantoin (nitrofurans) (300 µg), and doxycycline (tetracyclines) (30 µg). Additionally, a strip of the vancomycin (glycopeptide) E-test (BioMérieux; Marcy-l'Étoile, France) was applied to the inoculated Mueller Hinton agar (MHA) (Oxoid; Hampshire, UK) to determine vancomycin susceptibility of *S. aureus* and establish the minimum inhibitory concentration (MIC).

The MIC was determined following 16–18 hours of incubation at 37 °C [35], while for *Enterococci*, the antibiotic disks (Oxoid; Hampshire, UK) tested were: ampicillin (penicillin) (penicillins) (10 µg), vancomycin (glycopeptide) (30 µg), erythromycin (macrolide) (15 µg), tetracycline (tetracyclines) (30 µg), doxycycline (tetracyclines) (30 µg), chloramphenicol (phenicols) (30 µg), ciprofloxacin (fluoroquinolone) (5 µg), and nitrofurantoin (nitrofurans) (300 µg). Furthermore, *Enterococci* species that showed either intermediate or resistance by the vancomycin disk diffusion test were further investigated by vancomycin E-test strip (BioMérieux; Marcy-l'Étoile, France) based on the CLSI 2022 guideline [35].

In the disk-diffusion susceptibility test, antibiotic disks with known concentrations were placed on an agar plate, while the MIC was determined using an E-test strip. The colonies of a young culture were picked from the pure culture using a sterile wire loop and emulsified in 0.85% of normal saline to make bacterial suspension and compare with 0.5 McFarland turbidity standards (Thermo Fisher Scientific; Waltham, Massachusetts, USA) [35]. Using a sterile cotton swab, a lawn culture technique was used to inoculate evenly over the entire surface of Muller hinton agar (MHA) (Oxoid; Hampshire, UK) plates. Subsequently, the disk and vancomycin E-test strip (BioMérieux; Marcy-l'Étoile, France) were placed on the plates and left undisturbed for 2–3 minutes to allow the antibiotic to dissolve into the medium. The plates were then inverted and incubated at 37 °C for 18–24 hours. After overnight incubation, the inhibition zone diameter was measured with a ruler for the disk diffusion method, while the

starting point of the inhibition zone was determined for MIC. The outcomes were categorized as susceptible, intermediate, or resistant based on the CLSI 2022 guideline [35].

### Methicillin-resistant *Staphylococcus aureus* identification

To identify MRSA, pure young colonies of *S. aureus* isolates were inoculated on MHA (Oxoid; Hampshire, UK), and a cefoxitin (30 μg) disk was placed on the agar. The plates were incubated at 35 °C for 16 to 18 hours [15]. After incubation, the zone of inhibition was measured; isolates with an inhibition zone of ≥ 22 mm were classified as MSSA, while those with an inhibition zone of ≤ 21 mm were categorized as MRSA, in accordance with CLSI 2022 [35].

### Inducible clindamycin resistance identification

Inducible clindamycin resistance in *S. aureus* was determined using the D-test with erythromycin and clindamycin, as recommended by CLSI 2022 [35]. Mueller hinton agar (Oxoid; Hampshire, UK) was inoculated with a suspension of the isolate adjusted to a 0.5 McFarland standard (Thermo Fisher Scientific; Waltham, Massachusetts, USA). Erythromycin (15 μg) (Oxoid; Hampshire, UK) and clindamycin (2 μg) (Oxoid; Hampshire, UK) disks were placed 15 mm apart on the inoculated plates, which were then incubated at 37°C for 18 hours. Inducible clindamycin resistance was confirmed by the D-shaped flattening of the clindamycin inhibition zone around the erythromycin disk, indicating erythromycin-induced resistance [35].

### Operational definitions

**WBC count:** The normal WBC range was 4.9-14.3 x 109/L for patients aged between 1 to 5 years [36] and 3.7-14.3 x 109/L for patients between 6 to 14 years [37]. On the other hand, the normal WBC range was 3.1 - 10.6 x 109/L for patients ≥ 14 years old [38]. ANC: The normal ANC range was 1.9–7.5 x 109/L for patients aged between 1 and 5 years [39] and 0.90-8.71 x 109/L for patients aged between 6-14 years [37]. On the other hand, the normal ANC range was 1.4 - 7 x 109/L for patients ≥ 14 years old [38].

**Patients who had an upper respiratory tract infection:** patients with confirmed upper respiratory tract infection through culture or had signs and symptoms of upper respiratory infections and had undergone medication up to three months before the collection of the specimen.

### Quality control

The questionnaire was initially in English and then translated into the local language, Amharic. It was pre-tested on 5% of the estimated sample size at the Felege-Hiwot Compressive Specialized Hospital, Bahir Dar Oncology Treatment Center, to ensure its appropriateness and clarity. Clinical characteristics were obtained by reviewing the medical charts of each participant in the study. Sterility of the culture media was tested by incubating 5% of the prepared batch overnight at 37°C. Known standard reference bacterial strains, including *S. aureus* ATCC 25923, *E. faecalis* ATCC 12699, and *E. coli* ATCC 25922, were inoculated on selective culture media for performance testing.

### Data processing and analysis

Data were reviewed for completeness, cleaned, and double-entered into Epi Info 7, then analyzed using SPSS version 27. Frequency distributions and percentages described the socio-demographic and clinical characteristics of participants. The prevalence of MRSA and VRE was determined by the study population's proportions. Pearson Chi-Square analysis was performed as a preliminary analysis to assess the association between two categorical variables

at p < 0.05. Variables that met this threshold were further analyzed using crude and adjusted odds ratios through bivariable and multivariable logistic regression, respectively, with a 95% confidence interval. Variables with p < 0.2 in bivariable logistic regression were included in the multivariable analysis, with statistical significance defined as p < 0.05.

## Ethical consideration

Ethical clearance was obtained from the Ethical Review Committee of the School of Biomedical and Laboratory Sciences, College of Medicine and Health Sciences, University of Gondar, with reference number SBMLS/484, dated April 4, 2023. Written consent and assent were obtained from study participants and their guardians before specimen collection. All data was kept confidential during analysis using codes.

## Result

### Socio-demographic characteristics of study participants

In this study, 288 study participants were included. The proportion of male study participants was 51.0%, and 52.8% were aged between 41 and 60 years. The mean age of the study participants was 45.6 (SD ± 25.4) years (Table 1).

### Clinical characteristics of study participants

Different types of cancer cases were found, but breast, blood, and colon cancer collectively accounted for 112 (38.9%) of the cases. Data also showed that 52.1% of the study participants were receiving chemotherapy within 30 days prior to the study, and 21.2% of them had two cycles of chemotherapy. In addition, data also demonstrated that 16.3% of the study participants underwent major surgery and invasive procedures, within the last twelve months prior to the study. Furthermore, the majority of the study participants had normal WBC and normal ANC with percentages of 85.8% and 73.6%, respectively (Table 2).

### Prevalence of methicillin-resistant *Staphylococcus aureus* colonization

The overall prevalence of MRSA colonization among cancer patients was 11.1% (95% CI: 7.5, 14.7%; n = 32). Data from the current study showed that the anterior nares were the most common site for MRSA colonization. The proportion of MRSA isolated from the anterior nares and the GI tract was 23/32 (71.9%) and 9/32 (28.1%), respectively. Socio-demographic characteristics compared with MRSA colonization among cancer patients are summarized in Table 1. Twenty-two percent of the individuals with blood cancer and 20.0% of those with esophageal cancer were culture-positive for MRSA. In addition, 12.7% of the participants with Stage IV cancer were found to be culture-positive for MRSA. Cancer patients who had been receiving chemotherapy within 30 days prior to the study (15.3%) showed a relatively higher percentage of MRSA colonization. Study participants with a low WBC (35.7%) and low ANC (35.7%) range also showed a high proportion of MRSA colonization (Table 2).

### Prevalence of vancomycin-resistant *Enterococci* colonization

The overall percentage of culture-positive *Enterococci* species was 125 out of 288 (43.4%). The prevalence of VRE was 2.8% (95% CI: 0.9, 4.7; n = 8). Nasal colonization of *Enterococci* species was found in 8 out of 288 participants (2.8%), but no VRE was isolated from the anterior nares; all VRE cases were sourced from the gastrointestinal tract. Among male participants, VRE prevalence was 3.4%, and the highest proportion (3.3%) was observed in the 1–18-year age group. Vancomycin-resistant *Enterococci* proportions were 20.0% among participants with

**Table 1. Socio-demographic characteristics of the study participants and the distribution culture-positive methicillin-resistant *Staphylococcus aureus* and vancomycin-resistant *Enterococci* species at UoGCSH from May to July 30, 2023 (n = 288).**

| Participants characteristics | Categories | Frequency n (%) | MRSA n (%) | MSSA n (%) | p-value | VRE n (%) | Non VRE n (%) | p-value |
|---|---|---|---|---|---|---|---|---|
| Sex | Male | 147 (51.0) | 22 (14.1) | 21 (14.3) | 0.045* | 5 (3.4) | 61 (41.5) | 0.645 |
| | Female | 141 (49.0) | 10 (7.6) | 36 (25.5) | | 3 (2.1) | 55 (39.7) | |
| Age | 1-18 | 30 (10.4) | 1 (3.3) | 1 (3.3) | 0.242 | 1 (3.3) | 4 (13.3) | 0.743 |
| | 19-40 | 69 (24.0) | 8 (11.4) | 14 (20.0) | | 2 (2.9) | 28 (40.0) | |
| | 41-60 | 152 (52.8) | 16 (10.7) | 37 (24.7) | | 3 (3.0) | 71 (36.8) | |
| | >60 | 37 (12.8) | 7 (18.4) | 5 (13.2) | | 2 (5.3) | 14 (46.0) | |
| Residence | Urban | 157 (54.5) | 23 (14.6) | 26 (16.6) | 0.036* | 5 (3.2) | 51 (32.5) | 0.732 |
| | Rural | 131 (45.5) | 9 (6.8) | 31 (23.6) | | 3 (2.3) | 63 (48.1) | |
| Marital status | Single | 60 (20.8) | 5 (8.3) | 5 (8.3) | 0.161 | 3 (5.0) | 13 (21.7) | 0.149 |
| | Married | 191 (66.3) | 20 (10.5) | 42 (22.0) | | 3 (1.6) | 87 (45.5) | |
| | Divorced | 19 (6.6) | 2 (10.5) | 5 (26.3) | | 1 (5.3) | 9 (47.4) | |
| | Widowed | 18 (6.3) | 5 (27.8) | 5 (27.8) | | 1 (5.6) | 8 (44.4) | |
| Education status | No formal education | 183 (63.5) | 22 (12.0) | 28 (15.3) | 0.097 | 1 (0.5) | 75 (41.0) | 0.007* |
| | Primary education | 54 (18.8) | 6 (11.1) | 13 (24.1) | | 5 (9.3) | 20 (37.0) | |
| | Secondary education | 31 (10.8) | 2 (6.5) | 13 (41.9) | | 1 (3.2) | 12 (38.7) | |
| | College and above | 20 (6.9) | 1 (5.0) | 5 (20.0) | | 1 (5.0) | 10 (50.0) | |
| Occupation status | Government worker | 25 (8.7%) | 2 (8.0) | 7 (28.0) | 0.314 | 0 (3.7) | 11 (44.0) | 0.076 |
| | Private worker | 58 (20.1%) | 8 (13.8) | 11 (18.9) | | 2 (4.8) | 28 (48.3) | |
| | Housewife | 86 (29.9%) | 7 (8.1) | 20 (23.3) | | 1 (1.2) | 31 (36.0) | |
| | Farmer | 81 (28.1%) | 13 (16.0) | 17 (20.9) | | 4 (2.5) | 35 (43.2) | |
| | Other | 38 (13.2%) | 2 (5.3) | 2 (5.3) | | 1 (2.8) | 8 (21.1) | |

Legend: *The observed difference is statistically significant (p < 0.05);

Abbreviation: MSSA, methicillin-susceptible *Staphylococcus aureus*.

retinoblastoma and 3.3% for those receiving chemotherapy within the last 30 days prior to the study. Additionally, a higher proportion of VRE (10.6%) was noted among participants who had undergone invasive procedures in the 12 months prior to the study. VRE prevalence was 3.7% and 5.0% among cancer patients with elevated WBC and ANC, respectively (Table 2).

## Factors associated with colonization of methicillin-resistant *Staphylococcus aureus* among cancer patients

In the bivariable logistic regression analysis, socio-demographic and clinical characteristics such as sex, chemotherapy status, WBC, and ANC showed associations at p < 0.2, making them potential candidates for multivariate analysis. However, in the multivariable logistic regression, only low ANC was significantly associated with MRSA colonization among cancer patients (p < 0.05). Participants with low ANC levels (AOR = 13.050, 95% CI: 1.362–125.00, P = 0.026) were more likely to develop MRSA colonization compared to those with normal or high ANC levels (Table 3).

## Factors associated with colonization of vancomycin-resistant *Enterococci* among cancer patients

The study data indicated that cancer patients without formal education and those who had undergone an invasive procedure in the twelve months prior showed significant associations at p < 0.2 in binary logistic regression and qualified for multivariable analysis. However, the

**Table 2. Clinical characteristics of the study participants and the distribution culture-positive methicillin-resistant *Staphylococcus aureus* and vancomycin-resistant *Enterococci* species at UoGCSH from May to July 30, 2023 (n = 288).**

| Participants characteristics | Categories | Frequency n (%) | MRSA n (%) | MSSA n (%) | P value | VRE n (%) | Non VRE n (%) | P value |
|---|---|---|---|---|---|---|---|---|
| Type of CA | Blood | 41 (14.2) | 9 (22.0) | 5 (12.2) | 0.126 | 2 (4.9) | 14 (34.1) | 0.550 |
| | Colon | 30 (10.4) | 5 (16.7) | 5 (16.7) | | 3 (10.0) | 13 (43.3) | |
| | Breast | 41 (14.2) | 3 (7.3) | 16 (39.0) | | 0 | 20 (48.8) | |
| | Esophageal | 15 (5.2) | 3 (20.0) | 4 (26.7) | | 0 | 6 (40.0) | |
| | Lung | 12 (4.2) | 2 (16.7) | 4 (33.3) | | 0 | 7 (58.3) | |
| | Cervical | 18 (6.3) | 3 (16.7) | 1 (55.6) | | 1 (5.6) | 7 (38.9) | |
| | Pancreatic | 20 (6.9) | 2 (10.0) | 4 (20.0) | | 0 | 8 (40.0) | |
| | Wilms tumor | 10 (3.5) | 1 (10.0) | 1 (10.0) | | 0 | 1 (10.0) | |
| | Retinoblastoma | 5 (1.7) | 0 | 0 | | 1 (20.0) | 1 (20.0) | |
| | Other | 96 (33.3) | 4 (4.2) | 16 (16.7) | | 1 (1.0) | 40 (41.7) | |
| Cancer stage | Stage II | 65 (24.0) | 6 (9.2) | 16 (24.6) | 0.964 | 1 (1.5) | 31 (47.7) | 0.689 |
| | Stage III | 48 (16.7) | 6 (12.5) | 14 (29.2) | | 2 (4.2) | 25 (52.1) | |
| | Stage IV | 71 (24.7) | 9 (12.7) | 14 (19.7) | | 3 (4.2) | 34 (47.9) | |
| | In remission | 22 (7.6) | 2 (9.1) | 4 (18.2) | | 1 (4.5) | 5 (22.7) | |
| | not identified | 82 (28.5) | 9 (10.9) | 11 (13.4) | | 1 (1.2) | 25 (30.5) | |
| Underwent chemotherapy in the last 30 days prior to the study | Yes | 150 (52.1) | 23 (15.3) | 27 (18.0) | 0.017* | 5 (3.3) | 63 (42.0) | 0.550 |
| | No | 138 (47.9) | 9 (6.5) | 30 (21.7) | | 3 (2.2) | 54 (39.1) | |
| Chemotherapy Cycle | 1X | 34 (11.8) | 5 (14.7) | 6 (17.6) | 0.669 | 1 (2.9) | 8 (23.5) | 0.865 |
| | 2X | 61 (21.2) | 7 (11.5) | 11 (18.0) | | 3 (4.9) | 25 (40.9) | |
| | 3X | 38 (13.2) | 7 (18.4) | 9 (23.7) | | 1 (2.6) | 18 (47.4) | |
| | 4X | 19 (6.6) | 4 (21.1) | 3 (15.8) | | 0 | 7 (52.6) | |
| | >4X | 7 (2.4) | 1 (14.3) | 1 (14.3) | | 0 | 5 (71.4) | |
| | Not receiving | 129 (44.8) | 8 (6.2) | 28 (21.7) | | 3 (2.4) | 51 (39.5) | |
| Comorbidities | Yes | 46 (16.0) | 5 (10.9) | 8 (17.3) | 0.955 | 3 (6.5) | 19 (41.3) | 0.092 |
| | No | 242 (84.0) | 27 (11.2) | 49 (20.2) | | 5 (20.7) | 98 (40.5) | |
| Antibiotic use | Yes | 66 (22.9) | 6 (9.1) | 16 (24.2) | 0.660 | 2 (3.0) | 27 (43.9) | 0.887 |
| | No | 196 (77.1) | 26 (13.3) | 41 (20.9) | | 6 (2.7) | 90 (40.5) | |
| Length of hospitalization of patient | <48hrs | 224 (77.8) | 21 (9.4) | 46 (20.5) | 0.079 | 5 (1.8) | 92 (41.1) | 0.292 |
| | >48hrs | 64 (22.2) | 11 (17.2) | 14 (21.9) | | 3 (6.3) | 25 (39.1) | |
| Had major surgery done 12 month prior to the study | Yes | 47 (16.3) | 6 (12.8) | 15 (31.9) | 0.693 | 0 | 25 (53.2) | 0.205 |
| | No | 241 (83.7) | 26 (10.8) | 42 (17.4) | | 8 (3.3) | 92 (38.2) | |
| Had an invasive procedure done in the last 12 months prior to the study | Yes | 47 (16.3) | 6 (12.8) | 9 (19.1) | 0.693 | 5 (10.6) | 28 (59.6) | <0.001* |
| | No | 241 (83.7) | 26 (10.8) | 48 (19.9) | | 3 (1.2) | 97 (40.2) | |
| WBC report | Low | 14 (4.9) | 5 (35.7) | 1 (7.1) | 0.004* | 0 | 3 (21.4) | 0.783 |
| | Normal | 247 (85.8) | 22 (8.9) | 54 (21.9) | | 7 (2.8) | 107 (43.3) | |
| | High | 27 (9.4) | 5 (15.8) | 2 (7.4) | | 1 (3.7) | 7 (25.9) | |
| ANC report | Low | 56 (19.4) | 20 (35.7) | 10 (17.9) | <0.001* | 1 (1.8) | 26 (46.4) | 0.751 |
| | Normal | 212 (73.6) | 11 (5.2) | 42 (19.8) | | 6 (2.8) | 84 (39.6) | |
| | High | 20 (6.9) | 1 (5.0) | 5 (25.0) | | 1 (5.0) | 7 (35.0) | |

Legend: * The observed difference is statistically significant (p < 0.05); WBC: white blood count; ANC: absolute neutrophil count.

multivariable analysis revealed that only participants who underwent an invasive procedure in the past twelve months had a significant association with VRE colonization (p < 0.05). Specifically, these patients (AOR = 8.648, 95% CI: 1.870-39.992, P = 0.006) were more likely to develop VRE colonization compared to those without a history of such procedures (Table 4).

**Table 3. Bivariable and multivariable logistic regression analysis of factors associated with methicillin-resistant *Staphylococcus aureus* colonization among cancer patients admitted at UoGCSH, from May 1 to July 30, 2023.**

| Variables | Categories | MRSA n (%) | MSSA n (%) | COR (95% CI) | P value | AOR (95% CI) | P value |
|---|---|---|---|---|---|---|---|
| Sex | Male | 22 (14.1) | 21 (14.3) | 0.455 (0.207–0.998) | 0.049* | 2.273 (0.937–5.512) | 0.069 |
| | Female | 10 (7.6) | 36 (25.5) | 1(Ref) | | 1(Ref) | |
| Residence | Urban | 23 (14.6) | 26 (16.6) | 2.327 (1.036–5.223) | 0.041* | 1.911 (0.767–4.760) | 0.164 |
| | Rural | 9 (6.8) | 31 (23.6) | 1(Ref) | | | |
| Underwent chemotherapy in the last 30 days prior to the study. | Yes | 23 (15.3) | 27 (18.0) | 0.385 (0.172–0.865) | 0.021* | 1.750 (0.691–4.436) | 0.238 |
| | No | 9 (6.5) | 30 (21.7) | 1(Ref) | | 1(Ref) | |
| WBC | Low | 5 (35.7) | 1 (7.1) | 0.409 (0.095–1.765) | 0.231 | 1.160 (0.190–7.093) | 0.872 |
| | Normal | 22 (8.9) | 54 (21.9) | 2.324 (0.801–6.744) | 0.121* | 0.519 (0.135–1.992) | 0.339 |
| | High | 5 (15.8) | 2 (7.4) | 1(Ref) | | 1(Ref) | |
| ANC | Low | 20 (35.7) | 10 (17.9) | 0.095 (0.012–0.761) | 0.027* | 13.050 (1.362–125.00) | 0.026* |
| | Normal | 11 (5.2) | 42 (19.8) | 0.962 (0.118–7.857) | 0.971 | 0.152 (0.152–14.739) | 0.730 |
| | High | 1 (5.0) | 5 (25.0) | 1(Ref) | | 1(Ref) | |

Legend *The observed difference is statistically significant at (p.0.2) for bivariate and (p < 0.05) for multivariate.

Abbreviation: COR, crudes odds ratio; AOR: adjusted odds ratio; WBC: white blood count; ANC: absolute neutrophil count.

**Table 4. Bivariable and multivariable logistic regression analysis of factors associated with vancomycin-resistant *Enterococci* colonization among cancer patients admitted at UoGCSH, from May to July 30, 2023 (n = 288).**

| Variables | Categories | VRE n (%) | Non-VRE n (%) | COR (95% CI) | P value | AOR (95% CI) | P value |
|---|---|---|---|---|---|---|---|
| Education status | No formal education | 1 (0.5) | 75 (41.0) | 9.579 (0.576–159.392) | 0.115 | 0.157 (0.009–2.766) | 0.206 |
| | Primary | 5 (9.3) | 20 (37.0) | 0.516 (0.057–4.708) | 0.557 | 0.2701 (0.269–27.082) | 0.398 |
| | Secondary | 1 (3.2) | 12 (38.7) | 1.579 (0.093–26.776) | 0.752 | 0.975 (0.109–8.75) | 0.977 |
| | College and above | 1 (5.0) | 10 (50.0) | 1(Ref) | | 1(Ref) | |
| Had an invasive procedure done in the last 12 months prior to the study | Yes | 6 (12.8) | 15 (31.9) | 0.106 (0.024–0.460) | 0.003* | 8.648 (1.870–39.992) | 0.006* |
| | No | 26 (10.8) | 42 (17.4) | 1(Ref) | | 1(Ref) | |

Legend: *The observed difference is statistically significant at (p < 0.2) for bivariate and (p < 0.05) for multivariate;

Abbreviation: COR, crudes odds ratio; AOR: adjusted odds ratio

## Antibiotic resistance patterns of *Staphylococcus aureus*

Among the 89 *S. aureus* isolates analyzed, 32 were MRSA and 57 MSSA. The MRSA strains exhibited complete resistance to penicillin and high proportion of resistance to tetracycline (90.6%), doxycycline (84.4%), and trimethoprim/sulfamethoxazole (65.6%). In contrast, the MSSA isolates showed resistance rates of 80.7% to penicillin and 75.4% to tetracycline. Notably, neither MRSA nor MSSA demonstrated resistance to vancomycin. Inducible clindamycin resistance was identified in 12 of the isolates, resulting in a 13.5% positive percentage for the D-test

## Antibiotic resistance pattern of *Enterococci* species

Of the 125 *Enterococci* species isolates, 8 (6.4%) exhibited vancomycin resistance, with an MIC of 32 µg/ml. The VRE showed complete resistance to tetracycline and significant resistance to ampicillin (75%), erythromycin (87.5%), doxycycline (75%), and ciprofloxacin (75%). Conversely, VRE isolates demonstrated a higher susceptibility to nitrofurantoin (100%) and chloramphenicol (62.5%). Non-VRE species also showed high susceptibility rates of 98.3% to nitrofurantoin and 97.4% to chloramphenicol (Table 6).

## Multidrug-resistant patterns of *Staphylococcus aureus* and *Enterococci*

A total of 12 antibiotics from 11 classes (aminoglycosides, amphenicol, cephalosporins, fluoroquinolones, glycopeptides, lincosamides, macrolides, nitrofurans, penicillin, tetracyclines, and sulfonamides) were employed to evaluate MDR patterns of *S. aureus* isolates. The prevalence of MDR *Enterococci* species was 56.2% (n = 50), with resistance levels ranging from three to eight antibiotics (R3 to R8). Among the MDR *S. aureus*, 34.0% were resistant to three antibiotics and 30.0% to four. Conversely, 4.0% and 2.0% were resistant to seven and eight antibiotics, respectively (Table 7). The MDR *S. aureus* strains exhibited complete resistance to penicillin and high resistance to tetracycline (92.0%) and doxycycline (64.0%). However, these strains were also highly susceptible to chloramphenicol (94.0%), nitrofurantoin (78.0%), and gentamicin (80.0%) (Table 5).

**Table 5. Antibiotic susceptibility patterns of *Staphylococcus aureus*, and multi-drug resistant *Staphylococcus aureus* isolates among cancer patients at UoGCSH, from May to July 30, 2023 (n = 288).**

| Antibiotics | MRSA (n = 32) | | | MSSA(n = 57) | | | MDR (n = 50) | | |
|---|---|---|---|---|---|---|---|---|---|
| | S (%) | I (%) | R (%) | S (%) | I (%) | R (%) | S (%) | I (%) | R (%) |
| VAN | 17(53.1) | 15(46.9) | 0 | 48(84.2) | 9 (15.8) | 0 | 35 (70.0) | 15 (30.0) | 0 |
| FOX | 0 | – | 32 (100) | 57 (100) | – | 0 | 18 (36.0) | – | 32 (64.0) |
| CIP | 19 (59.4) | 4 (12.5) | 9 (28.1) | 43 (75.6) | 6 (10.5) | 8 (14.0) | 31 (62.0) | 4 (8.0) | 15 (30.0) |
| DOX | 5 (15.6) | 0 | 27 (84.4) | 28 (49.1) | 0 | 29 (50.9) | 8 (16.0) | 0 | 42 (64.0) |
| NIT | 24 (75.0) | 1 (3.1) | 7 (21.9) | 53 (93.0) | 2 (3.5) | 2 (3.5) | 39 (78.0) | 2 (4.0) | 9 (18.0) |
| ERY | 15 (46.9) | 1 (3.1) | 16 (50.0) | 49 (86.0) | 0 | 8 (14.0) | 29 (58.0) | 1 (2.0) | 20 (40.0) |
| CLY | 28 (87.5) | 0 | 4 (12.5) | 55 (96.5) | 0 | 2 (3.5) | 44 (88.0) | 0 | 6 (12.0) |
| TET | 3 (9.4) | 0 | 29(90.6) | 14 (24.6) | 0 | 43 (75.4) | 4 (8.0) | 0 | 46 (92.0) |
| SXT | 11(34.4) | 0 | 21(65.6) | 39 (68.4) | 1 (1.8) | 17 (29.8) | 19 (38.0) | 1 (2.0) | 30 (60.0) |
| CAF | 31(96.9) | 0 | 1 (3.1) | 54 (94.7) | 0 | 3 (5.3) | 47 (94.0) | 0 | 3 (6.0) |
| PEN | 0 | – | 32 (100) | 11(19.3) | – | 46(80.7) | 0 | – | 50 (100) |
| GEN | 26 (81.3) | 0 | 6 (18.7) | 53 (93.0) | 0 | 4 (7.0) | 40 (80.0) | 0 | 10 (20.0) |

Legend: PEN, penicillin; AMP, ampicillin; VAN, vancomycin; ERY, erythromycin; TET, tetracycline; CAF, chloramphenicol; NIT, nitrofurantoin; CIP, ciprofloxacin; SXT, Trimethoprim/ Sulfamethoxazole; GEN, Gentamycin; FOX, Cefoxitin; CLY, clindamycin; *S: sensitive, I: intermediate, R: resistant; (-) the symbol is used for antibiotics that have not been assigned a predetermined range according to CLSI 2022.

**Table 6. Antibiotics susceptibility patterns of *Enterococci* species and multi-drug-resistant *Enterococci* isolates among cancer patients at UoGCSH, from May 1 to July 30, 2023.**

| Drugs | Total VRE(n = 8) | | | Total Non VRE(n = 117) | | | MDR(n = 69) | | |
|---|---|---|---|---|---|---|---|---|---|
| | S (%) | I (%) | R (%) | S (%) | I (%) | R (%) | S (%) | I (%) | R (%) |
| VAN | 0 | 0 | 8 (100) | 108 (92.3) | 9 (7.7%) | 0 | 56 (81.1) | 5 (7.2) | 8 (11.6) |
| AMP | 2 (25.0) | – | 6 (75.0) | 39 (33.3) | – | 78 (66.7) | 9 (13.0) | – | 60 (87.0) |
| ERY | 1 (12.5) | 0 | 7 (87.5) | 34 (29.1) | 4 (3.4%) | 79 (67.5) | 9 (13.0) | 2 (2.9) | 58 (79.7) |
| DOX | 2 (25.0) | 0 | 6 (75.0) | 27 (66.7) | 0 | 90 (76.9) | 11 (15.9) | 0 | 58 (84.1) |
| CIP | 2 (25.0) | 0 | 6 (75.0) | 73 (62.4) | 5 (4.3) | 39 (33.3) | 25 (36.2) | 2 (2.9) | 42 (60.9) |
| TET | 0 | 0 | 8 (100) | 15 (12.8) | 0 | 102 (87.2) | 3 (4.3) | 0 | 66 (95.7) |
| CAF | 5 (62.5) | 0 | 3 (37.5) | 114 (97.4) | 1 (0.85) | 2 (1.7) | 63 (91.3) | 1 (1.4) | 5 (7.2) |
| NIT | 8 (100) | 0 | 0 | 115 (98.3) | 1 (0.85) | 0 | 68 (98.6) | 1 (1.4) | 0 |

Legend: AMP: ampicillin; VAN: vancomycin; ERY: erythromycin; TET: tetracycline; CAF: chloramphenicol; NIT: nitrofurantoin; CIP: ciprofloxacin; *S: sensitive, I: intermediate, R: resistant; (-) the symbol is used for antibiotics that have not been assigned a predetermined range according to CLSI 2022.

Eight antibiotics from seven classes (aminoglycosides, amphenicol, fluoroquinolones, penicillin, macrolides, tetracycline, and glycopeptides) were used to evaluate the MDR patterns of *Enterococci* species. The prevalence of MDR *Enterococci* species was 55.2% (n = 69), with resistance to ranging from 3 to 6 antibiotics. Specifically, 55.1% were resistant to three antibiotics, 34.8% to four, and 1.4% to six. Most MDR *Enterococci* species showed resistance to tetracycline (95.7%) and doxycycline (84.1%). In contrast, chloramphenicol, nitrofurantoin, and vancomycin exhibited greater sensitivity, with susceptibility rates of 91.3%, 98.6%, and 81.1%, respectively.

## Discussion

Colonization and subsequent infection by MRSA is a major global health issue. It is a serious issue for both the general public and certain patient populations, including cancer patients [40]. In the current study, the overall prevalence of MRSA colonization among cancer patients was 11.1% (95% CI: 7.5, 14.7%). Previous reports from different studies showed comparatively similar prevalence in with studies done in such as Egypt (9.2%) [41] and Ireland (11.5%) [4]. Nevertheless, a lower prevalence of MRSA was reported in Ethiopia (6.5%) [15], Iran (3.5%)

**Table 7. Multidrug-resistant pattern of *Staphylococcus aureus* and *Enterococci* species isolated from cancer patients UoGCSH, from May 1 to July 30, 2023 (N = 288).**

| Resistance rate | Frequency of MDR in *S. aureus* (n = 50) (%) | Frequency of MDR in Enterococci (n = 69) (%) | Combination of antibiotics for *S. aureus* | Combination of antibiotics for Enterococci |
|---|---|---|---|---|
| R3 | 17 (34.0%) | 38 (55.1%) | SXT,PEN.DOX, FOX,SXT,PEN TET,PEN,CN CLY, ERY,TET FOX,PEN,TET SXT,FOX,PEN FOX,PEN,DOX ERY,PEN,TET FOX,ERY,TET DOX,ERY,PEN SXT,TET,PEN CIP,TET,PEN | CIP,TET,AMP AMP,ERY,TET AMP,TET,CIP AMP,ERY,TET ERY,TET,CIP VAN,TET,AMP AMP,ERY,CIP |
| R4 | 15 (30.0%) | 24 (34.8%) | FOX,SXT,TET,PEN SXT,CLY,ERY,TET FOX,CIP,PEN,TET SXT,ERY,PEN,TET CIP,SXT,PEN,TET SXT,CAF,TET,PEN | AMP,ERY,CIP,TET ERY,CIP,CAF,TET AMP,ERY,TET,CAF |
| R5 | 12 (24.0%) | 6 (8.7%) | CAF,CIP,SXT,PEN.TET FOX,SXT,ERY.TET,PEN FOX,CIP,CLY,PEN.TET CIP,SXT,TET,GEN,PEN | VAN,AMP,ERY,TET,CIP VAN,ERY,TET,CIP,CAF |
| R6 | 3 (6.0%) | 1 (1.4%) | FOX,CIP,SXT,ERY,PEN,GEN CAF,SXT,ERY,CLY,TET,PEN | VAN,AMP,ERY,TET,CIP,CAF |
| R7 | 2 (4.0%) | 0 | FOX,SXT,ERY,TET,PEN,GEN,CIP | – |
| R8 | 1 (2.0%) | 0 | FOX,CIP,SXT,CLY,ERY,TET,PEN,GEN,TET | – |
| Not MDR | 39 | 56 | – | – |

Legend: AMP: ampicillin; CAF: chloramphenicol; CIP: ciprofloxacin; CLY: clindamycin; ERY: erythromycin; FOX, cefoxitin; GEN, gentamycin; NIT, nitrofurantoin; SXT: trimethoprim-sulfamethoxazole; TET: tetracycline; VAN: vancomycin.

[42], and the Czech Republic (4.0%) [43]. In contrast, a higher prevalence of MRSA colonization was observed in India (32%) [44] and Brazil (20.0%) [45]. Discrepancies in study results can arise from variations in study design, such as the effects of persistent versus intermittent colonization. Accurate characterization of types of colonization requires specimen collection at different times; however, this study only collected one specimen from a single period, which could lead to findings being classified as intermittent. To confirm persistent colonization, positive cultures must be obtained from the same individual at multiple time points.

Study participants with low ANC had the highest MRSA colonization with 35.7% having been colonized with MRSA. Additionally, a statistically significant association of Low ANC with MRSA colonization was reported (AOR = 13.050, 95% CI: 1.362-125.00, P = 0.026). The association between low ANC and MRSA colonization in cancer patients stems from the crucial role of neutrophils in the immune response. As key components of the innate immune system, neutrophils act as the first line of defense. However in patients undergoing chemotherapy the risk of neutropenia increases due to the targeting of rapidly dividing cells [46]. This condition results in a diminished capacity to mount an effective immune response, leaving patients vulnerable to infections [47]. When ANC levels fall below normal ranges, the body struggles to clear bacterial colonization, this vulnerability may lead to persistent colonization and an elevated risk of recurrent infections among these patients [48]. Even though chemotherapy is considered a risk factor in other studies for MRSA colonization [45], this association was not found in the current study. This could be attributed to variation in the study participants. Not all participants in this study received cancer chemotherapy, leading to a non-homogeneous population. This lack of homogeneity could have obscured the potential link between chemotherapy and MRSA colonization, making it difficult to draw definitive conclusions.

Antibiotic resistance in *S. aureus* has indeed become a significant global concern, as highlighted in multiple studies [25,40,49]. In this study, 95.5% of *S. aureus* isolates were susceptible to chloramphenicol; however, 100%, 80.9%, and 63% were resistant to penicillin, tetracycline, and doxycycline, respectively. This finding was in line with studies done in Ethiopia [15] and Kenya [49]. The widespread resistance to these essential antibiotics may be due to prevalent drug-resistant *S. aureus* strains in these areas, compounded by easy access to antibiotics over the counter and poor stewardship practices that contribute to misuse and heightened resistance. However the current study report is inconsistent with a study done in Pakistan [50] that reported 100% resistance to penicillin, 70% resistance to gentamycin, and 83% resistance to ciprofloxacin. A study in Oman [51] reported relatively higher antibiotic resistance to erythromycin (48%) and clindamycin (29%). A study in Germany [52] primarily reported isolates with high sensitivity to trimethoprim-sulfamethoxazole (92%) and doxycycline (88%). This disparity in the difference between the studies reported may be due to differences in antibiotic use and prescribing practices between different countries and locations, as well as differences in the quality of healthcare infrastructure and infection control practices.

In the current study, Inducible clindamycin resistance in *S. aureus* was identified in 13.5% of the isolated *S. aureus*. Previously, the prevalence of inducible clindamycin-resistant *S. aureus* was reported at 17.0% in Ethiopia [15] and 11.2% in Nigeria [53]. Study reports also showed that a higher prevalence of inducible clindamycin resistance among *S. aureus* was reported in Japan (31.9%) [54]. Clindamycin is often used to treat infections caused by *S. aureus*, including MRSA infections. However, the presence of inducible clindamycin resistance can worsen patient outcomes, as effective treatment may be delayed or inappropriate [55]. As a result incorporating the D-test into routine antimicrobial susceptibility testing can significantly enhance treatment outcomes for patients with Staphylococcus infection

The current study reported a 56.2% prevalence of MDR *S. aureus*, these MDR exhibited complete resistance to penicillin and a high proportion of drug resistance in antibiotics such as tetracycline (92.0%), doxycycline (64.0%), cefoxitin (64.0%), and trimethoprim-sulfamethoxazole (60.0%). The prevalence of MDR S. aureus varies in studies due to local resistance patterns, antibiotic use, geographic locations, and study periods. A recent Ethiopian study on *S. aureus* isolates in the Amhara region's four referral hospitals found a high MDR prevalence of 72.7% [56]. The emergence of drug resistance, especially to commonly used antibiotics like penicillin, tetracycline, doxycycline, cefoxitin, and trimethoprim-sulfamethoxazole, is a significant concern [56,57]. High rates of resistance to antibiotics such as penicillin and tetracycline serve as a clear indication of the negative outcomes that result from the overuse and improper administration of these medications, specifically in their role in fostering the development of antibiotic resistance within populations [15].

Colonization with VRE can lead to an increased risk of infection, which can lead to longer hospital stays and higher healthcare costs for cancer patients [24]. In the current study, 43.4% of the cancer patients were culture positive for *Enterococci* species, and the prevalence of VRE was 2.8% at 95% CI: 0.9, 4.7. Previous reports from India (2.9%) [58], Brazil (4.5%) [45] and Germany (1.4%) [59] showed similar findings. However, the current study's findings showed a lower prevalence than studies done in the Czech Republic (7%) [60] and Bulgaria (15.0%) [61]. The lower prevalence of VRE in the current study may be attributed to differences in detection methods, particularly molecular-based laboratory techniques that enhance detection sensitivity. Molecular techniques such as PCR allow for the detection of VRE strains that may not thrive in cultures or could be overlooked by standard culture methods, particularly in intricate specimens like fecal matter containing diverse enteric microorganisms that may impede the recovery of *Enterococci* species.

Data from the current study showed that *Enterococci* species showed high susceptibility to nitrofurantoin (100%), chloramphenicol (95.2%) and vancomycin (86.4%), However, high drug resistance was seen with antibiotics such as tetracycline (88.0%) and ampicillin (67.2%), which is consistent with previous study reports in Ethiopia [34] and India [62] and was inconsistent with a study done in Jimma, Ethiopia [63], that reported high resistance to ciprofloxacin (50%), erythromycin (63.2%), chloramphenicol (34.2%), and nitrofurantoin (32.4%) of the isolates. The current study reports were also inconsistent with the study done in Iran [8], that reported high resistance to vancomycin, chloramphenicol, and erythromycin. On the other hand, a study done in Pakistan [64] reported high resistance to chloramphenicol, ciprofloxacin, and tetracycline. The VRE isolates exhibited varying levels of resistance to different antibiotics such as tetracycline (100%), ampicillin (75.0%), erythromycin (87.5%), doxycycline (75.0%), and ciprofloxacin (75.0%). The reported result is parallel with a study done in Southern Thailand [65], that reported VRE isolates were 100% resistant to tetracycline, and 87.5% resistant to ampicillin and erythromycin. Additionally, a study from Egypt revealed that the pooled resistance rates among VRE isolates were 100% for tetracycline, 65.7% for ampicillin, 87.5% for erythromycin, and 75% for ciprofloxacin. Antibiotic resistance patterns exhibit significant geographical variation and evolve, highlighting the intricate nature of resistance development. Moreover, McDonnell et al. found a strong correlation between national antibiotic consumption rates and hospital antibiotic resistance rates, suggesting a link between usage and resistance levels across various bacterial strains [66].

The overall prevalence of MDR *Enterococci* species was 55.2%, consistent with a study in Hawassa, Ethiopia (54.6%). However, this contrasts with the higher prevalence rates found in previous studies in Gondar, Ethiopia (75.0%) (68) and Jimma, Ethiopia (89.5%) [67]. These variations in MDR rates among *Enterococci* species in different areas of Ethiopia show the need for continuous surveillance, antibiotic stewardship, and infection control to manage

the spread of drug-resistant pathogens and develop effective treatment strategies for varied resistance patterns in healthcare settings. Coincidentally, in the previous studies conducted in Ethiopia, the highest antibiotic resistance for MDR *Enterococci* species was found to be against tetracycline and beta-lactam antibiotics such as penicillin [33,34,63]. This aligns with the results of the current study on MDR *Enterococci* species: tetracycline (95.7%), ampicillin (87.0%). This consistency underscores the persistent challenge of antibiotic overuse despite the passage of time, indicating a lack of significant improvement in the prudent use of these antibiotics.

In the current study, multivariable regression analysis showed a statistically significant association between invasive procedures and infection caused by *Enterococci* species (AOR = 8.648, 95% CI: 1.870–39.992, P = 0.006). This finding is consistent with a study conducted in Turkey [68] and South Korea [18]. Invasive procedures involve accessing the body through incisions or insertions to perform diagnostic or therapeutic procedures [69]. During these invasive procedures, there is a risk of introducing bacteria from the environment or medical instruments into the body. While healthcare settings have strict infection control measures to minimize the risk of introducing bacteria, including VRE, it is still possible for bacteria to be introduced during these procedures.

This study initially hypothesized that the incidence of VRSA could rise due to the transfer of the vancomycin resistance gene from VRE to *S. aureus* in shared ecological niches [70], particularly given the high colonization risk in the study population. However, we found no VRSA strains, possibly due to the absence of co-colonization of MRSA and VRE among participants. Although our hypothesis was not supported, we reported the current prevalence of MRSA and VRE in this vulnerable population, highlighting the need for ongoing research into antibiotic resistance dynamics and comprehensive epidemiological studies to inform public health interventions.

## Limitations of the study

Due to the nature of this study being a cross-sectional study, it was difficult to determine whether participants had persistent or intermittent colonization. Moreover, the absence of molecular methods to characterize *Enterococci* species and assess the genotypic traits of antibiotic resistance patterns are significant limitations of the current study.

## Conclusion and recommendation

The study reveals a troubling prevalence of MRSA (11.1%) and VRE (2.8%) colonization among cancer patients, marking a significant public health issue for this vulnerable population. Importantly, a significant association was observed between MRSA colonization and low ANC, suggesting heightened susceptibility to bacterial infections due to weakened immunity. Moreover, the study also found a significant association between invasive procedures and VRE colonization. The findings underscore the risks faced by cancer patients, who are more likely to experience weakened immune systems from chemotherapy and undergo invasive procedures. The rising rates of antibiotic resistance in MRSA and VRE, particularly to penicillin and tetracycline, are concerning. This issue largely stems from easy access to these antibiotics without prescriptions, leading to their overuse and misuse in both healthcare and community settings, which significantly contributes to the development of antibiotic-resistant strains. The study also found 13.5% of *S. aureus* isolates exhibiting inducible clindamycin resistance, highlighting the need for D-testing during routine microbiology work. The study found a higher percentage of vancomycin-intermediate S. aureus colonization in this vulnerable population, which is concerning since vancomycin is currently the last treatment option

for severe MRSA infections in the area. The presence of these strains may complicate treatment and lead to failures, particularly in immunocompromised individuals, where increasing dosages is challenging due to the high nephrotoxicity associated with higher vancomycin levels. It is highly encouraged for MRSA nares screening as a potent antibiotic stewardship tool for MRSA infections known to have a high negative predictive value. Additionally, the potential for VRE colonization not only increases the risk of complications but also complicates treatment options, as currently vancomycin is the last drug of resort in the study area, leading to prolonged hospitalization and higher healthcare costs, which is why early identification of VRE colonization is necessary for this vulnerable population. Current findings offer preliminary insights, suggesting that results may improve with enhanced methodologies and larger sample sizes, including institution-based screening for MRSA and VRE while addressing the current study's limitations. Additionally, the study urges responsible parties to implement active surveillance systems to mitigate colonization risks in this vulnerable group.

## Supporting information

**S1 Fig. D-test positive (Inducible clindamycin resistance) methicillin resistant _Staphylococcus aureus_.** Legend: MRSA, methicillin resistant _Staphylococcus aureus_; MIC: minimum inhibition concentration; VISA: vancomycin intermediate _Staphylococcus aureus_.
(PDF)

**S2 Fig. _Enterococci_ species colony on Bile esculin agar**
(PDF)

**S3 Fig. VRE on muller hinton agar, MIC of 32μg/ml.** Legend: VRE, vancomycin resistant _Enterococci_; MIC: minimum inhibition concertation.
(PDF)

**S4 File. Minimal data set.**
(XLSX)

## Acknowledgments

We express our gratitude to the University of Gondar comprehensive and specialized hospital oncology treatment center for their cooperation during data collection. As well as to the Ethiopian Public Health Institute, Harar Regional Laboratory, and Felege Hiwot Specialized Hospital for their material aid. We also thank the study participants for their cooperation in providing the necessary specimens and information.

## Author contributions

**Conceptualization:** Eden Getaneh Mekonnen, Abebe Birhanu, Mucheye Gizachew, Baye Gelaw.

**Data curation:** Eden Getaneh Mekonnen.

**Formal analysis:** Eden Getaneh Mekonnen, Abebe Birhanu, Mucheye Gizachew, Baye Gelaw.

**Funding acquisition:** Eden Getaneh Mekonnen.

**Investigation:** Eden Getaneh Mekonnen, Mulugeta Yimer, Segenet Bizuneh, Mucheye Gizachew, Baye Gelaw.

**Methodology:** Eden Getaneh Mekonnen, Abebe Birhanu, Mucheye Gizachew, Baye Gelaw.

**Project administration:** Eden Getaneh Mekonnen, Mulugeta Yimer, Segenet Bizuneh, Mucheye Gizachew, Baye Gelaw.

**Resources:** Eden Getaneh Mekonnen.

**Software:** Eden Getaneh Mekonnen, Abebe Birhanu, Baye Gelaw.

**Supervision:** Abebe Birhanu, Mulugeta Yimer, Segenet Bizuneh, Mucheye Gizachew, Baye Gelaw.

**Validation:** Eden Getaneh Mekonnen.

**Visualization:** Eden Getaneh Mekonnen, Abebe Birhanu, Baye Gelaw.

**Writing – original draft:** Eden Getaneh Mekonnen, Abebe Birhanu, Baye Gelaw.

**Writing – review & editing:** Eden Getaneh Mekonnen, Abebe Birhanu, Mulugeta Yimer, Mucheye Gizachew, Baye Gelaw.

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
