## [Decision Letter · Decision Letter 0]

5 Aug 2024

PONE-D-24-24955Methicillin-resistant Staphylococcus aureus and Vancomycin-resistant Enterococci species colonization among cancer patients at the University of Gondar Comprehensives Specialized Hospital, Northwest EthiopiaPLOS ONE

Dear Dr. Mekonnen,

Thank you for submitting your manuscript to PLOS ONE. After careful consideration, we feel that it has merit but does not fully meet PLOS ONE’s publication criteria as it currently stands. Therefore, we invite you to submit a revised version of the manuscript that addresses the points raised during the review process.

We look forward to receiving your revised manuscript.

Kind regards,

Mohamed O Ahmed, Ph.D

Academic Editor

PLOS ONE

Journal Requirements:

Reviewers' comments:

Reviewer's Responses to Questions

**Comments to the Author**

1. Is the manuscript technically sound, and do the data support the conclusions?

Reviewer #1: Yes

Reviewer #2: Partly

Reviewer #3: Partly

2. Has the statistical analysis been performed appropriately and rigorously? 

Reviewer #1: Yes

Reviewer #2: Yes

Reviewer #3: No

3. Have the authors made all data underlying the findings in their manuscript fully available?

Reviewer #1: Yes

Reviewer #2: Yes

Reviewer #3: No

4. Is the manuscript presented in an intelligible fashion and written in standard English?

Reviewer #1: Yes

Reviewer #2: Yes

Reviewer #3: No

5. Review Comments to the Author

Reviewer #1: Reviewer Report

Comments

Firstly, I want to note that the discussion part has 7 pages and difficult to find the key message of the article. I think it should be rewritten having its key findings.

Abstract

Line 22-23 “However, the prevalence of these pathogens among cancer cases in Northwestern Ethiopia remains unreported.” I read some papers about methicillin resistance S.aureus in nasal swap in the study area. I think it needs revision like: However, the prevalence of………. under-reported.

Line 32: Are you sure that the incubation temperature of the stated organism is at 43-45°C?

Introduction

Line 54: “Cancer patients…..” already stated in line 53. I think you should replace it by “They..”

Methodology:

Line 205:” ………resistant based on the CLSI 2022 guideline”. Please cite the reference

Line 208 “……. incubated at 37 °C…” Are you sure that the incubation temperature for MRSA is 37°C? Please revise it.

Results:

Line 254: “….. 288 cancer study participants were included.” It requires rewriting. For example, ……Cancer patients were included or simply study participants…...

Discussion:

The discussion part has 7 pages and difficult to find the key message of the article. I think it should be rewritten having its key findings.

Line 381-383: “….Cancer treatments such as chemotherapy can weaken the body's ability to fight infections…..”. How do you know? How do you convince the reader? It is better to reference it to prove to the reader.

Line 412: “…, However,….” please write in lowercase or replace the comma by full stop before it.

Line 437: “…participants however,…” please put ; before however, or full-stop then begin with uppercase However….

Line 439: by Debre Markos…. I think, it's better to say in Debre Markos



Conclusion:

Based on your findings, any key message was not forwarded to the readers and the community. I advise the authors to add the recommendation in the conclusion part.

Reviewer #2: The current study is very interesting; but it needs a major revision and the authors should address the following comments to improve the quality of the manuscript:

-Please write the scientific names of bacterial pathogens in the correct form all over themanuscript and in the References section (should be italic).

-The manuscript should be revised for grammar mistakes.

-Please modify the title. Include associated factors on it

- is it enough mannitol salt agar and bile esculin agar to identify those spcies????????????????

- on the abstract part you must include 95 % CI for each point prevalence of your findings and also please include AOR with p-value for the associated factors, and add more details about the used methods and most prevalent results in the abstract.

-Introduction: (This part needs more work and synthesis of ideas):

-added reference on line 90 and 91 about definations of Multi-drug resistance (MDR).

-Give a hint about the virulence factors and the mechanism of disease occurrence of pathogens.

- The authors should illustrate the public health importance concerning the magnitude, mortality, morbidity in drtails.

Material and methods:

- On line 104 delete A cross-sectional institution-based study and replaced by An institution-based cross-sectional study

- On sample size determinations, are you sure is not recent study on your study title, rechecked it. And recalculate it ???????

- How to stratified your study participants and include number of study participants in each starata

- Why you exclude patients who had an upper respiratory tract infection or who showed signs of coughing or difficulty breathing within the last three consecutive months, and patients who did not have updated whole blood cell reports were excluded from the study.

- Rewrite Specimen collection and transportation In the correct way

- Specific references should be added to all the used methods and techniques.

- Discuss in detail the methods of isolation and identification of bacterial pathogens. Besides, specific references should be added.

- Specific references should be added to all the used methods and techniques.

- Add the company, city, and country of the used bacterial media and reagents that were used in the biochemical identification of isolates.

- Antimicrobial susceptibility testing:

- Add the names of the antimicrobial classes of the tested antibiotics.

- PCR-based detection of antibiotic resistance genes in the recovered isolates should be carried out.

- Results: rewrite again focus on major finding

- Discuss in detail the phenotypic characters of the isolates.

- The detection of antimicrobial resistance genes in the retrieved isolates should be carried out if applicable.

- The correlation between the phenotypic and genotypic MDR should be performed,

- Discussion:

- The authors are advised to illustrate the real impact of their findings without repetition of results.

- Illustrate the different mechanisms of antimicrobial resistance in Methicillin-resistant Staphylococcus aureus and Vancomycin-resistant Enterococci species

-Conclusion Should be rephrased to be sounded. A real conclusion should focus on the question or claim you articulated in your study.

Reviewer #3: Tittle: Methicillin-resistant Staphylococcus aureus and Vancomycin-resistant Enterococci species colonization among cancer patients at the University of Gondar Comprehensives Specialized Hospital, Northwest Ethiopia

Comments

The authors in this paper have taken an attempt to report the Methicillin-resistant Staphylococcus aureus and Vancomycin-resistant Enterococci species colonization among cancer patients at the University Of Gondar Comprehensives Specialized Hospital, Northwest Ethiopia This is of course is very interesting subject which creates interest among readers. However the paper seems very poor standard and not at par for international readers which must be revised before publication based on the following comments

Major comments:

1. I suggest the tittle should be corrected as follows “colonization rate of Methicillin-resistant Staphylococcus aureus and Vancomycin-resistant Enterococci species among cancer patients at the University of Gondar Comprehensives Specialized Hospital, Northwestern Ethiopia”.

2. The authors should be remove all author information (email addresses) except cross ponding authors from the cover page.

3. In abstract section:

-what was the biochemical test used to identify the S. aureus and Enterococci species? Please include this in method section.

-Line 32: it says: Nasal swab and fecal specimens were collected and inoculated onto mannitol salt agar and bile esculin agar and incubated at 37°C and 43-45°C. Is the incubate has range or fixed value, please use fixed numerical value either 43, 44 or 45 °C Instead of using range value.

-line 34-37: It says: “To detect methicillin-resistant Staphylococcus aureus, a Cefoxitin disk was used. Inducible clindamycin resistance in Staphylococcus aureus was determined by the D test. Epi-info version 7 and Statistical Package for the Social Sciences version 27 were used for data entry and data analysis”. Please correct as follows: To detect methicillin-resistant Staphylococcus aureus, a Cefoxitin disk was used. Inducible clindamycin resistance in Staphylococcus aureus was determined by the D test. Epi-info version 7 and Statistical Package for the Social Sciences version 27 were used for data entry and data analysis, respectively.”

-line 44-47: the authors should be including the adjusted odds ratio with p-value.

- The authors should be including the recommendation in abstract section.

4. Introduction section:

-the authors should include the magnitude of antimicrobial resistance of S. aureus, Enterococci species, and MRSA in Ethiopia especially in the study area.

5. Materials and methods:

- Line 108-111: it says: “The UoGCSH is one of the largest tertiary referral and teaching hospitals in the region, serving over 13 million residents in northwestern Ethiopia. Among other services, the established cancer treatment center provides services for cancer cases. The cancer treatment center has outpatient and inpatient departments with 40 beds each in pediatric and adult oncology”. –the author should be provide the source of this information or reference is needed.

- The Source population and Study population seems to the same. The authors should be stated clearly.

- Sample size determination correction formula should be indicated to know your sample size is 288.

- Inclusion and exclusion criteria not clearly stated, is author only used the study participants newly diagnoses CA patients? Or included the CA patients in treatment stage?

-sampling technique should be convent sampling technique. If you are using stratified systematic random sampling, Please show us in figure how you are going to use it.

-why authors did not identify Enterococci species at species level?

- Why authors did not identify VRSA?

- In Antibiotic susceptibility test section: the authors should be mentioned the antimicrobial disk manufacturer company and country.

-line 205: put reference here

-For Methicillin-resistant Staphylococcus aureus identification: the authors should be include interpretation how to categorized MRSA OR MSSA

-The authors should be rewrite line 211-213.

-Ethical consideration: -please state the protocol number, date, month, and year.

-what was the fate of the positive individuals, are they treated or not?

6. Results section:

- Subheading should correct as follows “Socio-demographic characteristics of study participants” for Socio-demographic characteristics and “Clinical characteristics study participants” for Clinical characteristics.

- The presentation of MDR in Table 5 and Table 6 does not show the real MDR. So, the authors should be reanalysis MDR result as R1, R2, and R3…….or by listing the antimicrobials. Generally the authors do not analyze MDR isolates.

- Table 6 should be revised for example percentage is not indicated

7. Discussion section:

- The discussion should be improved with clear justification for example at end of discussion line 529- 533.

8. Limitations of the study:

-The authors should be put the limitations of the study before conclusion.

9. Conclusion:

-The conclusion is very shallow, so, please improve it.

10. The authors should be write recommendation. Research without recommendation is unusual.

11. There is old reference like 2008, so, please use the update one.

- Finally there are topographical, editorial, and grammatical errors. Please rewrite

6. PLOS authors have the option to publish the peer review history of their article (what does this mean? ). If published, this will include your full peer review and any attached files.

**Do you want your identity to be public for this peer review?** For information about this choice, including consent withdrawal, please see our Privacy Policy .

Reviewer #1: No

Reviewer #2: No

Reviewer #3: No

---

## [Author Response · Author response to Decision Letter 1]

3 Oct 2024

Thank you for your valuable feedback; we have worked to enhance the manuscript's quality based on your insightful comments. We have carefully considered each point raised and made substantial revisions to improve clarity and coherence. We believe these changes not only address your concerns but also elevate the overall narrative and argumentation of the manuscript. We appreciate your guidance during this process and hope that the revisions meet your expectations. Thank you once again for your time and consideration.

---

## [Decision Letter · Decision Letter 1]

1 Nov 2024

PONE-D-24-24955R1Colonization of methicillin-resistant Staphylococcus aureus  and vancomycin-resistant Enterococci  and its associated factors in cancer patients at the University of Gondar Comprehensive Specialized Hospital, Northwest EthiopiaPLOS ONE

Dear Dr. Mekonnen,

Thank you for submitting your manuscript to PLOS ONE. After careful consideration, we feel that it has merit but does not fully meet PLOS ONE’s publication criteria as it currently stands. Therefore, we invite you to submit a revised version of the manuscript that addresses the points raised during the review process.

We look forward to receiving your revised manuscript.

Kind regards,

Mohamed O Ahmed, Ph.D

Academic Editor

PLOS ONE

Reviewers' comments:

Reviewer's Responses to Questions

**Comments to the Author**

1. If the authors have adequately addressed your comments raised in a previous round of review and you feel that this manuscript is now acceptable for publication, you may indicate that here to bypass the “Comments to the Author” section, enter your conflict of interest statement in the “Confidential to Editor” section, and submit your "Accept" recommendation.

Reviewer #2: All comments have been addressed

Reviewer #3: All comments have been addressed

2. Is the manuscript technically sound, and do the data support the conclusions?

Reviewer #2: Yes

Reviewer #3: Yes

3. Has the statistical analysis been performed appropriately and rigorously? 

Reviewer #2: Yes

Reviewer #3: Yes

4. Have the authors made all data underlying the findings in their manuscript fully available?

Reviewer #2: Yes

Reviewer #3: Yes

5. Is the manuscript presented in an intelligible fashion and written in standard English?

Reviewer #2: Yes

Reviewer #3: Yes

6. Review Comments to the Author

Reviewer #2: Reviewer comments to authors in PLOSONE

Reviewer: This manuscript is markedly improved from the last time I reviewed it. I thank the authors for their efforts and especially for the improved. I continue to have some concerns about the data that should be addressed in the methods and limitations sections of the manuscript.

On abstract: please include the gaps rather than prevalence also include others

- L---21 delete at the University of Gondar Comprehensive Specialized Hospital b/s you stated it on the objective part, however include your sample size, sampling technique, and populations

- L---22 added pretest on structured questionnaires

- L—23 include method and way of transportations of your sample collections

- Can directly inoculate the specimens in to manitol salt agar and Bile esculin agar after you collect the sample

- Pearson Chi-Square, bivariable and multivariable logistic regression analyses identified associations, with a p-value < 0.05 considered statistically significant. Rewrite again it

- When you use S. aurous first write the full name, then you must use the abbreviations all over the documents and the same true for enterococcus to its consistence

Introductions: it seem correct

Methods and materials

- In study setting include more about flow of cancer patients

- L----135 remove study periods b/c you already mission on study setting

- It’s better to put the figure rather than word to explain your sampling procedure to indicate proportional allocations and indicates the number of participants from each strata.

- What is the difference b/n your study populations and inclusions criteria???? Is it similar??????

- L--- 171-172 they have redundancy idea please rewrite it?

- L –173- 176 it is better to move it in to data quality control part

- Generally you have to use pretested structured questioners to collect the data so, where you have gotten the questionnaire, and how to develop it and give different reference for it????

- L--- 273-277 it needs modifications, please see in details.

Result and dissections

- It written in appropriate way but Write major finding for your result since it is very wide? And also discussions part

- I would like some discussion of their findings against past developments.

- You are very much aware of the limitations of the study.

- Though it discovered in general nothing not known for the local situation it will be a good starting point to act.

- I suggest the authors to also add some references of data about other regions where those bacterial infections are endemic: Middle Est and Asia (India and South Est Asia). A comparison between Ethiopia and those regions can make the study more interesting and relevant.

- Finally, there are topographical, editorial, and grammatical errors. Please rewrite

- The conclusion is very shallow, so, please improve it.

Reviewer #3: All comments have been addressed by the authors, Hence, I suggested that this manuscript is accepted for publications.

7. PLOS authors have the option to publish the peer review history of their article (what does this mean? ). If published, this will include your full peer review and any attached files.

**Do you want your identity to be public for this peer review?** For information about this choice, including consent withdrawal, please see our Privacy Policy .

Reviewer #2: No

Reviewer #3: No

---

## [Author Response · Author response to Decision Letter 2]

26 Dec 2024

We sincerely appreciate your valuable feedback, as it has played an instrumental role in our efforts to enhance the overall quality of the manuscript. Your insights and suggestions have significantly contributed to our understanding and have guided us in making meaningful improvements.

---

## [Editor Report · Decision Letter 2]

14 Jan 2025

Colonization of methicillin-resistant Staphylococcus aureus  and vancomycin-resistant Enterococci  and its associated factors in cancer patients at the University of Gondar Comprehensive Specialized Hospital, Northwest Ethiopia

PONE-D-24-24955R2

Dear Dr. Eden,

We’re pleased to inform you that your manuscript has been judged scientifically suitable for publication and will be formally accepted for publication once it meets all outstanding technical requirements.

Kind regards,

Mohamed O Ahmed, Ph.D

Academic Editor

PLOS ONE

---

## [Editor Report · Acceptance letter]

PONE-D-24-24955R2

PLOS ONE

Dear Dr. Mekonnen,

I'm pleased to inform you that your manuscript has been deemed suitable for publication in PLOS ONE. Congratulations! Your manuscript is now being handed over to our production team.

Kind regards,

on behalf of

Dr. Mohamed O Ahmed

Academic Editor

PLOS ONE